# The Emperor’s New Clothes—An Epistemological Critique of Traditional Chinese Veterinary Acupuncture

**DOI:** 10.3390/ani9040168

**Published:** 2019-04-15

**Authors:** Manuel Magalhães-Sant’Ana

**Affiliations:** 1CIISA—Centre for Interdisciplinary Research in Animal Health, Faculty of Veterinary Medicine, University of Lisbon, 1300-477 Lisboa, Portugal; mdsantana@gmail.com; 2Ordem dos Médicos Veterinários, Av. Filipe Folque, 10J, 4º Dto., 1050-113 Lisboa, Portugal

**Keywords:** veterinary acupuncture, bloodletting, humoralism, placebo, Traditional Chinese Medicine, evidence-based medicine, Thomas Kuhn, history of science

## Abstract

**Simple Summary:**

Complementary and alternative medicines have gained increased popularity in the veterinary field. Among them, Traditional Chinese Veterinary Medicine, including acupuncture, has emerged as one of the main alternatives to conventional veterinary medicine. This paper relies upon an epistemological approach to investigate conceptual, historical and scientific assertions about veterinary acupuncture made by their advocates. Argument by analogy is used to demonstrate that Traditional Chinese Veterinary Medicine is based on pre-scientific principles, similar to those of humoral medicine and bloodletting, and that acupuncture is, in effect, a placebo. The paper concludes with recommendations for veterinary regulators and colleagues.

**Abstract:**

Within the last few decades, complementary and alternative medicines have gained increased popularity in the veterinary field. Although many authors have exposed the scientific fallacies and historical misconceptions used to justify such therapies, those efforts have not succeeded in detracting veterinary practitioners from embracing them. Notably, Traditional Chinese Veterinary Medicine (TCVM), including acupuncture, has emerged as one of the main alternatives to conventional veterinary medicine. In this paper, analogical reasoning is used to investigate conceptual, historical and scientific assertions made by the advocates of TCVM. The paper is divided into two parts: The first aims to appraise conceptual and historical claims made by veterinary acupuncturists. I defend that TCVM is a pre-scientific construct, similar to humoral doctrine, and that acupuncture is analogous to bloodletting. The second part is focused on scientific evidence of clinical application of acupuncture in the dog, showing how science is yet to validate veterinary acupuncture and defending that claims of efficacy are due to placebo effect. It is suggested that veterinary acupuncture needs to abandon Traditional Chinese Medicine and embrace science-based medicine *tout court*. On the other hand, high quality scientific studies, including randomized controlled trials, need to be presented. Veterinary regulators must bring the issue of non-conventional therapies into their agendas.

## 1. Introduction

Within the last few decades, complementary and alternative medicines (hereafter, non-conventional therapies) have gained increased popularity in the veterinary field. Although several textbooks [1,2] and internet scientific fora (such as *The Skeptvet* [3]) have exposed the scientific fallacies and historical misconceptions used to justify such therapies, those efforts have not succeeded in detracting veterinary practitioners from embracing them. Notably, Traditional Chinese Veterinary Medicine (TCVM), including acupuncture, has emerged as one of the main alternatives to conventional veterinary medicine. As a way of example, in 2018 alone, two professional organisations devoted to veterinary acupuncture have been established in Portugal.

This paper relies upon an epistemological approach to the issues. In particular, argument by analogy is used to investigate conceptual, historical and scientific assertions made by the advocates of TCVM. Analogical reasoning is considered one of the chief inductive epistemic approaches used in science [4], and argument by analogy has been described as “an explicit representation of a form of analogical reasoning that cites accepted similarities between two systems to support the conclusion that some further similarity exists” [5]. The focus is on acupuncture—the insertion of needles through the skin at specific points for diagnostic or therapeutic purposes—and not on Traditional Chinese Herbal Medicine.

The paper is divided into two parts: The first consists of an appraisal of conceptual and historical claims made by veterinary acupuncturists, which will illustrate why TCVM is a pre-scientific construct, similar to humoral doctrine, and how acupuncture is analogous to bloodletting. The second part is focused on scientific evidence, using the clinical application of acupuncture in the dog as case study to elucidate why science is yet to validate the clinical value of veterinary acupuncture, and make the case that claims of efficacy are due to placebo effect. The paper concludes with recommendations for veterinary regulators and colleagues.

## 2. Conceptual and Historical Critique 

Acupuncture is arguably the most reputed therapeutic approach resultant from Traditional Chinese Medicine (TCM). The Yellow Emperor Huang Ti, the last of the legendary Celestial Emperors, is widely considered the founding father TCM at around 2700 BCE [6]. Huang Ti is a mythological figure, a Taoist deity, who is credited for having created not only Chinese medicine, but also writing, astronomy, the calendar, the wheel and many others [6,7,8]. In turn, Taoism is theosophy, a religious philosophy centred around the concept of *Tao*, the Way of Nature, spanning China and other Asiatic societies for centuries [9]. 

Throughout the ages, the Taoist principles of life have been incorporated into TCM, which have also been adapted to the veterinary field. Klide and Kung [10], in their textbook *Veterinary Acupuncture*, provide a detailed account of the principles of TCVM. These principles have remained virtually unchanged since the book was first published in 1977 and have guided veterinary acupuncturists ever since. Recently, in an article published for the Portuguese veterinary community by a non-conventional veterinarian, we are told that:
**AP** [acupuncture]’s theory of action is grounded on **TCM**, on the concepts of **Yin** and **Yang**, that represent the balance, of **Qi**, that represents the vital **energy**, and on how these **energies** move, how they flow and distribute throughout the body through **meridians** that connect the various organs and regions of the body. The **AP** points are located in the skin and mostly in the path of the **meridians** and through their stimulation it is possible to alter the **energy** flow and interfere with the functioning of the organs. These **Eastern** concepts, which were developed from the observation of nature, from biological phenomena and from responses produced by organisms to environmental stimuli, support the **TCM** theory and are validated by the functioning of **AP**.”(p. 36, translation and emphasis mine) [11]

The Taoist philosophy behind acupuncture and TCM/TCVM practiced in the West draws from the assumptions that a so-called vital energy *Qi* [Ch’i] flows throughout the body within meridian pathways, manifesting itself in the *yin* and *yang* principles, which can be accessed and modulated via points in the skin called acupoints. These concepts may seem novel and alien to contemporary Western veterinarians who stare with awe and fascination into this previously ignored thousand-year-old philosophy of medicine. But were these words written one hundred and fifty years ago they would have caused little surprise. Up until the late nineteenth century, veterinarians would have believed that in order to cure an organic disorder you would have to access specific points in the skin and perform bloodletting, a therapy reminiscent of a system known as humoral doctrine. The sixth edition of *A Compendium of Cattle Medicine (…)*, dated from 1842, includes bleeding as one of the recommended operations, together with drenches, clysters and setons. In this book we are instructed that:
In all inflammatory disorders bleeding is of the first importance, and cannot be performed too early. (…) One copious bleeding, that is, until the pulse sinks, will frequently crush the disorder at once; (…) From one to two galleons of blood may generally be taken from a heifer or steer, or even from a milch cow.(p. 304) [12]

As late as 1897, bleeding of horses was still being recommended by some of the most prestigious veterinary surgeons [13]. For the sake of the argument, let us consider that I wanted to convince the reader of the benefits of bleeding or bloodletting (if not yet convinced by the quote above). Grounded on analogical reasoning, I would use the following rationale:
**Bloodletting**’s theory of action is grounded on **humoral doctrine**, on the concepts of **cold**/**wet** and **hot**/**dry**, that represent the balance, of **Pneuma**, that represents the vital **spirit**, and on how these spirits move, how they flow, and distribute throughout the body through vessels that connect the various organs and regions of the body. The **bloodletting** points are located in the skin and mostly in the path of the vessels and through their stimulation it is possible to alter the **spirit** flow and interfere with the functioning of the organs. These **Western** concepts, which were developed from the observation of nature, from biological phenomena and from responses produced by organisms to environmental stimuli, support the **humoral doctrine** theory and are validated by the functioning of **bloodletting**.

Although these arguments are perhaps not convincing today, they were guiding principles for veterinarians and animal healers for hundreds of years: Faced with a febrile horse covered in sweat (i.e. hot and wet), they would diagnose an imbalance of the vital spirit *Pneuma*, caused by a *dyscrasia* (i.e. excess) of blood, an imbalance that could only be treated with the removal of the excess blood through cuts in the skin near the vessels. I can only sympathise with the credulity of my veterinary colleagues after a session of ‘copious bleeding’ succeeded in having the horse’s body temperature lowered and the sweat disappear (i.e. turning cold and dry), meaning that the blood had returned to *eucrasia* (i.e. equilibrium) and the humoral imbalance had been successfully reverted, allowing the vital force *Pneuma* to flow unimpeded throughout the body. It was the observation of nature that supported bloodletting and it was the apparent functioning of bloodletting that validated the humoral doctrine.

The above exercise of *argument by analogy*—in which allegedly Eastern keywords were replaced by their Western counterparts—works to demonstrate that humoral veterinary medicine and TCM share a common philosophical background. They are two faces of the same coin (and would a coin have three faces, we could also include Ayurvedic medicine) [14]. Similarly, acupuncture and bloodletting are essentially variants of the same therapy that, as we shall see, have adapted to different cultural and religious environments.

Humoral medicine and TCM are not based on scientific evidence, but instead followed pre-scientific cosmogonic concepts that have been around for thousands of years, at least since the time of Greek philosopher Empedocles (c.500–430 BCE) [8] and Chinese philosopher Confucius (551–479 BCE) [9]. The four humours (blood, phlegm, yellow bile and black bile) and the five elements (wood, fire, earth, metal and air) are vitalist concepts used to help us make sense of overwhelmingly complex entities, such as health, disease and death. The body is a microcosm, obeying to the same natural laws of the universe [2,8,15]. *Pneuma* and *Qi* represent the same animist vital force, holding everything together in harmony.

Fortunately, physicians and veterinarians have long abandoned bloodletting, and I wish not to revive it. The humoral doctrine was replaced by Pasteur’s germ theory and by Virchow’s cellular pathology during the second half of the nineteenth century, and thus casting the medical tradition based upon the authority of Hippocrates, Aristotle and Galen aside [16]. Conversely, acupuncture, for both humans and animals, has never received so much attention and TCM has become a global multimillion-euro business, supported by the World Health Organization for reasons other than scientific [17].

At this point, the reader might be struggling to find similarities between bloodletting and acupuncture, as the former was invasive and aggressive, while the latter seems delicate, almost innocuous. Acupuncture, however, was not always about fine stainless-steel needles and in the words of the historian Epler Jr. “bears little resemblance to its ancestral version” [18] (p. 337). Again, analogical reasoning proves useful: A closer look into the nine ancient acupuncture needles (Figure 1a), leads us to conclude that most of them would be betted described as lancets, similar to those used by European barber-surgeons (Figure 1b), rather than needles: With flat surfaces, sharp edges and geometrical points, they would have been employed for lancing, firing and bloodletting, instead of needling [15,18,19]. Other needles are similar to surgical probes and scoops used by Roman physicians for cauterizing and probing wounds and fistulae (Figure 1c).

Others have pointed to the fact that the word *zhen*, often translated as needle, can refer to any sharp object used for cauterization, lancing, bleeding and surgery [15,19,20]. The term *jing luo* [經絡], which has become known in the West as meridian—a word coined by the French diplomat George Soulié de Morant (1878–1955)—used to designate a system of channels or vessels containing both blood and *Qi* (or *Pneuma*) [18,19]. Additional clues on what is meant by needling can be found in the classic acupuncture text, *Huang Ti Nei Ching Su Wen* [The Yellow Emperor’s Inner Classic of Medicine], considered by some to be from the third century BCE, but partially written as late as the ninth century AD [20]:
(A disease of) the minor-yang (vessel) causes the abdominal region to be painful as if the skin was being pricked by a pin. Initially (the patient) is not able to bend down, nor then to look up, and then he cannot twist around. Needle the minor-yang (vessel) at the end of the Ch’eng bone so that blood flows out. (…)(p. 351) [18]

It is then likely that what has become known as needles were originally lancets aimed at reaching the vascular system. In the first Korean veterinary treatise, the *Sin pyeon jip seong ma ui bang* [New compilation of prescriptions for horse diseases] dated from 1399, we are told that: “Needling on the six Maek, and letting blood flow [from the acupuncture point], is the very prescription by Shen Nong” (sic) [21]. This strongly suggests that, at least in late fourteenth century Korea, needling was being used for phlebotomy, bearing in mind that the Korean book is a compilation of Chinese hippiatric texts. But as medical practice evolved, and technology became available, the lancets became progressively thinner and smaller, leading to the dry needles known today.

But why have Chinese lancets evolved to needles, and not European ones? The social and religious context of Imperial China can help explain this evolutionary design. One of the basic tenets of Taoism consisted of the *wu-wei*, i.e. the contemplation of nature with ‘no active intervention’ [9] (p. 101). Furthermore, the sacred nature of the human body, one of the philosophical foundations of Confucianism (a humanistic religious doctrine introduced by Confucius), prevented surgeries and anatomical dissections [6] (p. 3). In contrast, the Catholic Church has never prohibited surgery or therapeutic bloodletting [8] (p. 144) and has probably never officially banned human anatomic dissections [22]. As a consequence, medical surgery in Imperial China (including bloodletting and cauterisation) was considered of lesser value than herbal medicine [9,15], and maybe even proscribed. In line with this argument, it is noteworthy that no such thing as Traditional Chinese Surgery has withstood the test of time. One of the few Chinese surgeons whose name reached us, Hua-To (c.190–265 AD), was allegedly charged with the attempted murder of emperor Tsao-Tsao when trying to apply trepanation to cure his migraines, and consequently executed [8] (p. 82).

And this is where acupuncture comes in. Filiform needle acupuncture would provide a minimally invasive alternative to the body wrecking practices of bloodletting, trepanation or surgery in general, without violating religious and social norms. The main therapeutic purpose of needling the vessels is turned towards restoring the equilibrium of *Qi*, whereas the letting of blood becomes progressively irrelevant [18]. In an era when often the best treatment was no treatment at all, as the prescribed ‘treatments’ often resulted in the worsening of clinical signs or even death, opting for acupuncture, despite its therapeutic inefficiency, offered nonetheless a much less invasive and dangerous alternative.

It is far from settled when acupuncture—resembling anything like contemporary needling—was established and reliable evidence capable of defending its antiquity has proved elusive. Veterinary acupuncture is often claimed to have been practiced for thousands of years [10,23,24,25]. These claims, which have been refuted by others [20,26], are implausible and stem from tradition rather than facts. Heaven, Earth and Man were all considered to be governed by the *Tao*, but there is no indication that the Tao of Man was believed to be applicable to animals. Samantha Scott highlights that animals would be considered as soulless beings, rendering the concepts of circulating *Qi* and meridians irrelevant [27].

Despite these epistemological weaknesses, veterinary acupuncturists insist that figures depicting acupunctural charts in animals can be traced back to hundreds of years ago [10,24,25]. Again, the evidence proves deceptive. One of such claims regards the *Yuan Heng liao ma ji* [Yuan and Heng’s Collection for Treating Horses], from 1608 AD [25] (p. 8). It has been shown that this book does not include any reference to acupuncture and that the charts are indeed used to indicate the anatomic location of pathologic conditions or points for cauterization and bleeding [20,26]. For example, Figure 2a is known to illustrate how hair whorls were used in divination [20], with some of the captions surrounding the horse describing its temperament (choleric, calm, sad, moribund). These terms are equivalent to the names used in mediaeval times to describe the relative abundance of each humour: Sanguine (blood), phlegmatic (phlegm), melancholic (black bile), and choleric (yellow bile) [8] (p. 99). Furthermore, the resemblances of these illustrations to those from European veterinary textbooks describing the topographic anatomical distribution of diseases of the horse are striking (Figure 2b), especially when comparing the horses’ flank and croup (Figure 2c). These illustrations were used to encapsulate the most relevant medical knowledge in one single image, and there is no reason to believe that Chinese veterinary textbooks were any different.

Imrie and colleagues [20], after thoroughly examining ancient Chinese veterinary texts, hypothesise that veterinary acupuncture is a relatively recent invention rather than a traditional Chinese practice. In summary, no historical evidence exists to support the use of therapeutic needling in animals before the second millennium AD, and even then references are dubious [20]. We do know that by the nineteenth century, acupuncture would have been no stranger to European veterinarians. Veterinary ‘acupuncturisation’ was most likely first introduced in France. It was the editor of the French scientific journal *Recueil de Médecine Vétérinaire* [Compendium of Veterinary Medicine], Prof. Girard fils [son] who, excited by experiments in humans, encouraged veterinary practitioners to apply acupuncture in animals. [28] p. 172. On March 7th 1825, the veterinarian Bouley jeune [Jr.] reports three observations of dry needle acupuncture in two mares suffering from claudication and in a draught horse with paralysis, with little or no success [29], in what can be claimed as the first confirmed description of veterinary acupuncture in Europe. Experiments, such as these must have continued because Gourdon’s *Éléments de Chirurgie Vétérinaire* [Essentials of Veterinary Surgery], published in 1857, dedicates a full chapter to acupuncture. Gourdon concludes that:
*Tested by them* [early veterinary acupuncturists] *almost immediately, it produced quite disparate results: some favorable, others null. Failures were the most numerous; what can be attributed, in part, to the inadequate application made, from the very beginning, of this means.*(p. 2, translation mine) [30]

Gourdon also claimed that acupuncture was rarely used in isolation, and that results would improve if electricity was applied to the needles (what was called *galvano-puncture*), a practice that will be discussed in further detail in the next section. Much like other pre-scientific practices that proved mostly ineffective, veterinary acupuncture was abandoned (with few exceptions, cf. [10]) until being again resurrected by the new age movement during the 1970s.

The endorsing of Traditional Chinese Medicine by the People’s Republic of China during the latter decades of the twentieth century has disseminated its practices throughout the world and led Western researchers to use modern science to legitimise traditional acupuncture. Contemporary veterinary acupuncturists have sought to ‘squaring the circle’ and explain the effects of acupuncture using modern neurophysiology, without rejecting TCVM and the concepts of *yin*, *yang*, *Qi* and meridians [31,32,33]. For example, Cantwell clarifies that:
this medicine [TCVM] uses a metaphoric language to describe the pathophysiology of disease and patterns of treatment. The traditional concept surrounds qi (pronounced chee), which is usually translated as energy or life force. The qi circulates through all parts of the body via pathways called meridians. Up to 350 points along and around these meridians have increased bioactivity and are called acupuncture points.(p. 53) [31]

Cantwell starts by suggesting that the concept of *Qi* is only metaphorical, but ends up by trying to justify its existence through acupuncture points (acupoints) and their alleged ‘increased bioactivity’, a scientifically fuzzy concept that has never been demonstrated [34,35,36]. Acupuncturists have claimed that acupoints coincide with nerve endings, blood vessels, tendons and many others [31,32]. However, the same can be said about non-acupoint anatomic regions, which helps explain why there is no consensus regarding the number and precise anatomic location of so-called acupoints, especially in animals [37].

Here we reach a point where the *epistemes* of science-based medicine and TCM cannot be easily reconciled. Relying upon scientific concepts with the view of rendering the principles of TCM valid not only seems a spurious exercise, but also undermines veterinary science. This antagonism has been brilliantly espoused by the philosopher-physicist Thomas Kuhn in his book *The Structure of Scientific Revolutions* [38]. Following Kuhn, scientific breakthroughs and medical discoveries have resulted in a ‘paradigm shift’, in which the new paradigm (science-based medicine) is not only different, but ‘better’ than the old one (vitalism, either via Humoralism or TCM). Science-based medicine and vitalism are ‘incommensurable’ because a ‘scientific revolution’ has replaced one paradigm by another. In the words of Thomas Kuhn,
“once it has achieved the status of paradigm, a scientific theory is declared invalid only if an alternate candidate is available to take its place. (…) The decision to reject one paradigm is always simultaneously the decision to accept another, and the judgment leading to that decision involves the comparison of both paradigms with nature and with each other. (…) to reject one paradigm without simultaneously substituting another is to reject science itself.”(pp. 77–79) [38]

In other words, it is not possible to embrace modern veterinary science and simultaneously follow the principles of TCVM because TCVM is antithetical to scientific thought: While science is based on hypotheses that can be tested, TCVM is based on dogmas that must not be abjured. Science is a systematized body of knowledge, whereas TCVM an immutable belief system. Scientific knowledge is hence gathered, interpreted and continuously validated—or re-evaluated—in light of existing evidence, whereas TCVM is merely aimed at justifying its pre-conceived ideologies. 

The Taoist principles of life are a fundamental component of Chinese culture and have also greatly influenced Eastern societies. Henceforth, its societal role is not to be contested. However, the principles of TCM, just like humoralism, should have no place in modern veterinary medicine. Since humoral medicine is part of a distant past, a ‘collective amnesia’ prevents us from seeing the resemblances between *Qi* and *Pneuma* and between acupuncture and bloodletting [15]. Furthermore, the allure of exotic ‘medical orientalism’, together with an opportunity to differentiate oneself from others, have however driven many veterinary colleagues towards embracing TCVM. New age consumerism and the pursuit for everything ‘natural’ or ‘traditional’ has attracted clients to TCVM and veterinarians have responded by providing those services [15]. But if humoral medicine has been abandoned, why is TCVM still alive (and kicking)? Surely acupuncture is no quackery. That is why we need to look closely at the scientific evidence of veterinary acupuncture.

## 3. Scientific Critique

While acupuncture has been applied to both domestic and wild animal species (including non-human primates, birds, reptiles and fish [39], which use lies outside the scope of this paper), this reflection focuses primarily on prominent examples from different areas of veterinary acupuncture in dogs, where most of the clinical interventions have been performed, and for which more evidence has been reported. Case-studies illustrating the range of claims of efficacy of acupuncture in dogs are henceforth presented, including for epilepsy, anaesthesia, analgesia, and rehabilitation medicine, followed by an argument by analogy.

There is abundant literature detailing the neurophysiological effects of acupuncture in terms of sympathetic (*Yang* equivalent) and parasympathetic (*Yin* equivalent) activity (e.g. [40]). No one would dispute the fact that inserting a needle into a living organism induces a physiological response, a response that can be analysed and measured. There is, however, a significant difference between explaining the neurophysiological effects of needle insertion and attributing it putative therapeutic effects. The Cochrane Collaboration has recently compiled over 60 systematic reviews regarding the efficacy of human acupuncture and concluded that there is currently no evidence to suggest that acupuncture can be useful for treating any disease or disorder in humans [41]. Similarly, a 2006 systematic review of the veterinary literature found “no compelling evidence to recommend or reject acupuncture for any condition in domestic animals” based on studies considered of low average methodologic quality [42].

Arguably one of the studies that may have contributed to the credibility of veterinary acupuncture stated that gold wire acupuncture implants were beneficial in controlling canine epilepsy [43]. This study was originally aimed at comparing the electroencephalographic recordings of dogs suffering from uncontrolled seizures before and after gold wire acupuncture was applied (under anaesthesia). Although no difference was found in that respect, experimental animals showed significantly fewer seizures with reduced severity after treatment. Despite being published in a reputable journal, the study failed to provide reliable evidence: The dog population was heterogenous and uncontrolled, dogs’ underlying pathology was poorly described and, most importantly, the study overlooked the anticonvulsant effect of propofol, a third-line drug for controlling *status epilepticus* in dogs [44] (p. 429). After ten years, to the best of my knowledge, these results have never been replicated.

Although it would be expected that the quality of evidence would have improved in the last decade, evidence concerning veterinary acupuncture has remained of low average quality. A study aiming at comparing the sedative effects of dexmedetomidine in dogs found that the Governing Vessel 20 (GV20) acupuncture point may achieve a higher intensity and prolonged duration of sedation when compared with routine intramuscular injection [45]. However, the study fails to consider that results may be due to differences in drug route administration (subcutaneous injection on the head versus gluteus muscle injection) instead of any relevant effects of acupuncture. A pilot study of gut mobility of 6 dogs found tenuous and transitory effects on the use of dry needle acupuncture when compared to a control [46]. Researchers found that acupuncture prolonged, rather than decreased, gastric emptying times for a short time period. However, such small effect in such a sample size precludes any definitive conclusions and may suggest that results may be nothing more than spurious.

Veterinary acupuncture has also gained a reputation in rehabilitation therapy. One of its most reputed approaches is electroacupuncture, the successor of nineteenth century *galvano-puncture*. Some have claimed that electroacupuncture is more effective than decompressive surgery in improving neurologic signs of dogs with intervertebral disk disease [47]. This prospective clinical trial compared dogs subjected to decompressive surgery, electroacupuncture or both. The authors acknowledge that 7 out of the 10 dogs in the decompressive surgery group had hemilaminectomy alone whereas both hemilaminectomy and disk fenestration was performed in all 11 dogs of the mixed therapy group, rendering any comparison problematic. Moreover, the authors have not considered that the efficacy of electroacupuncture may be better explained by electrical nerve stimulation rather than by acupuncture itself [48]. More invaluable information would have emerged by comparing the efficacy of electroacupuncture with that of Transcutaneous Electrical Nerve Stimulation (TENS), a procedure with similar neurochemical mechanisms and transduction pathways. A study published in PLOS One, found that TENS combined with acupuncture was effective at regulating the organic effects of controlled hypotension in dogs [49]. However, there is no indication that results would have been any different if TENS had been used on its own. The overlap between the two therapies seems deliberate, since the authors have avoided any reference to acupuncture in the title and abstract.

Concerns raised by studies, such as these must necessarily bring to the fore the question of whether acupuncture is no more than a placebo. The placebo effect is not a prerogative of human beings and, contrary to common understanding, can also be found in animals [50,51]. In particular, the placebo effect can emerge in veterinary rehabilitation medicine which relies upon subjective assessments and where treatments may be perceived as more effective than they actually are by owners and veterinarians [52,53]. But is there any concrete evidence that veterinary acupuncture may work as a placebo?

One of the most notorious claims of success of veterinary acupuncture is in treating dog hip dysplasia with gold bead implants [54,55], although such assertions are not shared by others [56]. A study by Hielm-Bjorkman and colleagues [57] found clinical improvements in dogs with hip dysplasia after gold wire acupuncture, but did not find a difference between treatment and control (14 G needle insertion with no implantation of gold wire at locations which were not acupoints). This indicates that sham acupuncture is indistinguishable from true acupuncture, rendering any training in meridians and acupoints irrelevant. A post-mortem examination (immunohistochemistry and histology) of dogs with spontaneous osteoarthritis treated with periarticular pure gold bead implants demonstrates that its therapeutic effects can be explained by the release of inflammatory factors associated with needle insertion [58] and not by any changes in a supposed *Qi* or energy flow. In other words, gold bead implant acupuncture does not act upon osteoarthritis, but instead works by masking its clinical signs (*via* neuromodulation) under the veil of periarticular inflammation. Consequently, the discussion lies in pondering whether introducing 14 G needles in the periarticular space—even without implants—in order to cause chronic tissue inflammation may be an acceptable treatment option for dogs with osteoarthritis [37] and not whether traditional Chinese veterinary acupuncture may have therapeutic effects. Finally, the potential complications caused by the embedding of irretrievable gold fragments should not be neglected [37].

Still with regard to dog hip dysplasia, a randomised, controlled clinical trial aimed to assess the efficacy of dry needle acupuncture versus a control anti-inflammatory drug (oral carprofen) and a placebo group (oral placebo drug with no acupuncture) [59]. The study posited that acupuncture was as effective as carprofen in controlling pain, leading the authors to conclude that “acupuncture performed with the protocol and acupoints used in this study appears to be a viable option for improving the quality of life of dogs with HD [hip dysplasia]”. However, these conclusions are highly misleading, since neither acupuncture nor carprofen were significantly better than placebo, and it was shown that the authors had resorted to ‘cherry-picking’ and inappropriate statistical analyses to support their claims [60,61]. Moreover, the efficacy of the acupuncture protocol (and acupoints) that was used could only be tested had it been compared against a sham protocol (using non-acupoints), which was not the case. Despite the exposure, the authors were unwilling to retract their conclusions, supported on their ‘clinical impressions’ of the benefits of acupuncture [60,61].

Another prominent use of acupuncture regards the effects of the Renzhong GV26 acupoint for neonatal resuscitation and anaesthetic recovery of animals [23,62,63,64]. Similar anaesthetic effects have recently been reported in humans [65]. Needing of the nasal philtrum is thought to stimulate respiration by activating neural afferent A and C fibres. This extraordinary effect merits further research and would, at first, urge us to consider that the potential therapeutic effects of needling *per se* should not be neglected. However, any clinical effects caused by the painful stimulation of an anatomical region compatible with GV26 acupoint are insufficient to demonstrate that the *Du Mai* meridian is a tangible entity or that the GV26 acupoint is in any way meaningful. In effect, to my knowledge, no study in animals has compared the effects of needling of the GV26 acupoint with that of a sham control (proximate non-acupoint). In truth, these effects have little to do with TCVM and go to show how contemporary Western acupuncturists have come to consider that every therapy which may involve inserting needles in an animal’s body as a form of acupuncture. This reflection compels us to consider yet another analogy: Claiming that the principles of TCVM are the reason why needling of the nasal philtrum may have clinical effects is no more valid than claiming that the resolution of pneumothorax by virtue of inserting a 14 G needle in the chest—which may coincidently be at an acupoint—is proof of the therapeutic value of acupuncture. 

In conclusion, claims of success of acupuncture can be explained by more rational, and arguably simpler, scientific explanations that sustain the claim that acupuncture is a ‘theatrical placebo’ [65]. But in addition to the empirical evidence, analysing the scientific representations of acupuncture is also useful. One of the leading references in the field of veterinary physiotherapy includes acupuncture as a nonpharmacological approach to treating chronic pain [66]. In this textbook we are told that:
**Needles** and **needle**-induced changes are believed to activate the built-in survival mechanisms that normalize homeostasis and promote self-healing. In this context, **acupuncture** can be defined as a physiologic therapy coordinated by the brain that responds to the stimulation of manual and electrical **needling** of peripheral sensory nerves, in which **acupuncture** does not treat any particular pathologic system, but normalizes physiologic homeostasis and promotes self-healing(p. 249, emphasis mine) [66]

All in all, this is a rather vague definition, and its validity warrants further scrutiny. We are told that acupuncture does not treat any particular pathologic system, but it is perhaps odd that the authors may conceive that acupuncture can induce homeostasis without acting upon any particular physiological body system, considering that pathologies affect body systems. Since homeostasis presupposes a balance between body systems, it can only be understood if its underlying physiological mechanisms are established, which is not the case for acupuncture. Moreover, the authors rely on the concept of ‘self-healing’ to justify acupuncture’s physiological effects. Although self-healing is a term commonly used in materials science (referring to materials capable of regenerating, often for medical purposes) it is not an accepted term in medical science, being absent in contemporary human or veterinary medical dictionaries known to the author. Besides, it resonates with the ‘mind cure’ movements of the first half of the twentieth century [67,68].

Reference to self-healing can only be plausibly understood in two ways: Either cure ensued as a result of the natural course of illness or due to the placebo effect (a third possible explanation—some kind of metaphysical healing, such as a miracle—has not been considered for obvious reasons). More importantly, in any case, either form of self-healing is independent of any putative therapeutic action of acupuncture and would occur in the absence of the therapy or in the presence of a similar—yet sham—form of treatment.

In other words, the authors are acknowledging one of two things: Either acupuncture produces no harm, and hence does not interfere with the body’s ability to heal itself or, at best, it works as a placebo. In effect, in another exercise of argument by analogy, should we replace the word acupuncture by the word placebo, *mutatis mutandis*, the sentence would read without having its meaning changed:
**Placebo** and **placebo**-induced changes are believed to activate the built-in survival mechanisms that normalize homeostasis and promote self-healing. In this context, **placebo** can be defined as a physiologic therapy coordinated by the brain that responds to the stimulation of [ ] sensory nerves, in which **placebo** does not treat any particular pathologic system, but normalizes physiologic homeostasis and promotes self-healing.


This is arguably a sound definition of placebo and works as an example of how acupuncturists, in an attempt to fill the void between therapy and effect, can fall into the epistemological trap of considering acupuncture for what it really is: A body-brain physiologic therapy that can promote health and well-being, i.e. a placebo [69].

## 4. Final Recommendations

This paper takes a critical look at Traditional Chinese Veterinary Acupuncture, supported by epistemological arguments. Harriet Hall (MD) once said that “science and critical thinking don’t come naturally to humans. Evolution shaped us to prefer stories to studies, anecdotes to analyses, emotion to evidence” [70], and three suggestions can be drawn from this reflection. If veterinary acupuncture is to reach the status of a conceptually respectable therapeutic approach, it is up to their advocates to abandon the pre-scientific vitalism of Traditional Chinese Medicine and embrace scientific method *tout court*, a view shared, to some extent, by some acupuncturists [71]. This would also mean accepting to only apply acupuncture in cases for which there is solid scientific evidence, a message that European scientific veterinary organisations need to convey to their members. On the other hand, it would also demand more high quality scientific studies, including randomized controlled trials, since the efficacy of acupuncture—or lack thereof—can only be reliably assessed with improved research design (with adequately sized and homogeneous samples, proper controls, blinding, and randomization) in order to test the most common clinical uses of veterinary acupuncture [72]. Failure to do so, would represent an unacceptable waste of time and resources. 

Veterinary acupuncture is a growing therapeutic field. Dry needle acupuncture remains the most recognised form of acupuncture, although it is becoming ‘old fashioned’ when compared to seemingly more sophisticated acupuncture practices that have emerged in recent years. In addition to the already mentioned electroacupuncture and gold bead implants, veterinary acupuncture now includes such techniques as hemo-acupuncture, laser-acupuncture, aqua-acupuncture, pneumo-acupuncture, bee venom acupuncture (apipuncture), ozone-acupuncture, stem cell acupuncture and probably many others. These approaches have not been fully tested by reliable clinical trials and are yet to be validated by science. However, they are currently being presented to veterinary clients across the world as if they were accepted and established veterinary therapies. This can endanger animal welfare and the public good and poses a serious reputational risk to the veterinary profession [2].

Such context demands that veterinary regulators bring the issue of non-conventional therapies into their agendas, including which restrictive measures should be imposed on them. As a way of example, the Portuguese and Spanish medical regulators have produced a consensus statement condemning pseudo-therapies and pseudo-sciences in human medicine, although no names were mentioned [73]. In comparison, the Royal College of Veterinary Surgeons (UK) position statement on complementary and alternative medicines is arguably softer and mainly focused on homeopathy [74]. The Ordem dos Médicos Veterinários [Portuguese Veterinary Order] has recently endorsed a working group to appraise, upon request, the scientific merits of non-conventional therapies [75]. In addition, the issue of which professionals are competent in administering these therapies must also be resolved. In a UK survey, 30% of horse owners report that acupuncture was being administered on their horses by lay people, despite only vets being legally allowed to administer acupuncture to animals [76].

Thomas Kuhn has cautioned that “because scientists are reasonable men, one or another argument will ultimately persuade many of them. But there is no single argument that can or should persuade them all” [38] (p. 158). With these words in mind, I do not expect to change the mind-set of those colleagues who have invested years and resources in acquiring knowledge and skills in TCVM. But I do hope that these arguments can reach those who have recently embraced TCVM, as well as those who are contemplating it. I urge them to dissect the evidence with a sceptical scalpel, whether it be conceptual, historical or scientific claims. I also urge them to follow evidence-based medicine, in order to build a self-conscious and responsible opinion regarding the usefulness of acupuncture in veterinary practice.

## Figures and Tables

**Figure 1 animals-09-00168-f001:**
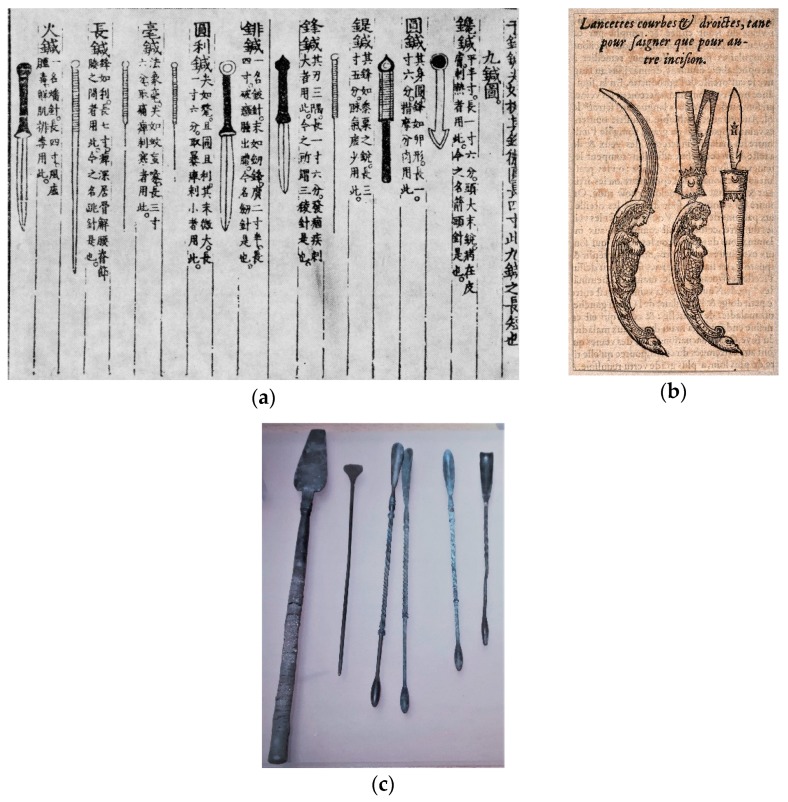
Analogies between ancient Chinese ‘needles’ and European surgical instruments. (**a**) The nine ancient needles of acupuncture and their clinical applications according to Yang Jizhou [杨继洲]. *Fac-simile* from *Zhen Jiu Da Cheng* [针灸大成, Great Compendium of Acupuncture and Moxibustion], China, 1601. Credits unknown. (**b**) Three lancets for use in the incision and bleeding procedures: Two lancets have ornamental handles in the grotesque style, while another is shown stored in a sheath. Woodcut, 1560/1600. Source: Wellcome Collection. Licence CC BY (https://wellcomecollection.org/works/wuguqbkh). (**c**) From left to right: Cauterizing instrument, probe and scoop-probes. Roman Empire, 1st–2nd Century AD. Bronze. Museu da Farmácia [Pharmacy Museum], Lisbon, Portugal. Author’s photo.

**Figure 2 animals-09-00168-f002:**
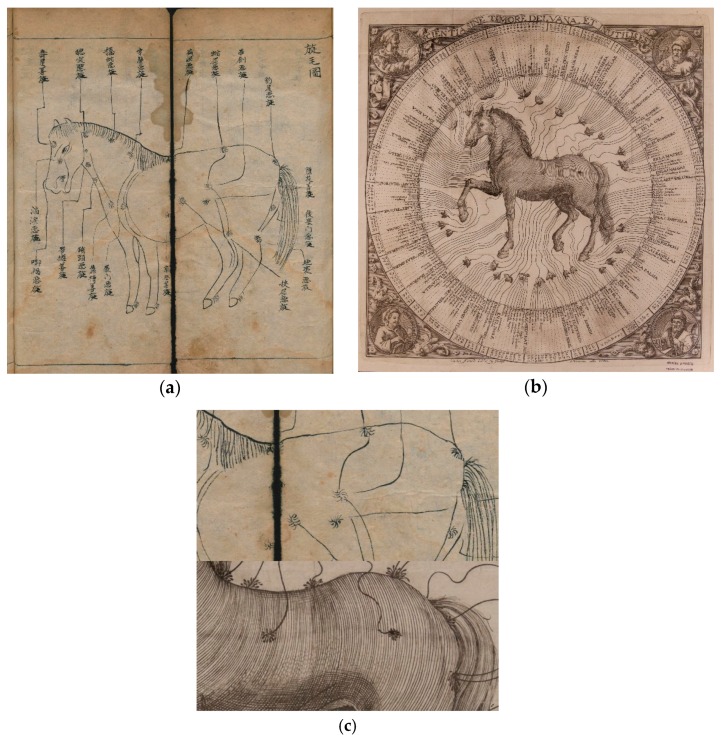
Analogies between ancient Chinese and European hippiatry treatises, with two examples of topographic descriptions of the horse. (**a**) Early 20th Century handwritten copy of the *Yuan Heng liao ma ji* [纂圖元亨療馬集] (Author: Yu Benyuan [喻本元], China, 1608). Source: Staatsbibliothek zu Berlin—PK, Unschuld collection (Slg. Unschuld 8963). Licence: CC by-nc-sa 3.0 de (http://resolver.staatsbibliothek-berlin.de/SBB0001CAB100000000). (**b**) Woodcut from *Sanidad del cavallo y otros animales sujetos al arte de albeyteria: ilustrada con el de herrar* (Author: Salvador Montó y Roca; Ed. Joseph Estevan Dolz, Valencia, Spain, 1742). Source: BIVALDI, Biblioteca Valenciana Nicolau Primitiu, BNP (http://bivaldi.gva.es/es/consulta/registro.cmd?id=10020). (**c**) Detail from both illustrations comparing the horses’ flank and croup.

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
