# Peer review of "The Emperor’s New Clothes—An Epistemological Critique of Traditional Chinese Veterinary Acupuncture"

_animals, 2019, doi:10.3390/ani9040168_

Round 1
Reviewer 1 Report
As a proponent of the use of EBVM and as a veterinarian concerned with the animal welfare and ethical issues associated with the use of CAVM I was pleased to read this paper that takes a critical and sceptical look at the use of veterinary acupuncture. I thought the paper achieved its aim and arguments were made in a logical and appropriate fashion. The paper maintained my interest.
The strength of this paper is that it adds to the literature on this topic and examines the basis of the belief in Traditional Chinese Veterinary Acupuncture and brings together various threads, historical and scientific regarding the use of acupuncture in animals. The arguments that Traditional Chinese Veterinary Medicine is based on pre-scientific principles and acupuncture is a placebo are well made and the reference list is extensive.
The paper is well structured and easy to read although some of the English needs tidying up.
Although these views have been made many times before in textbooks, published papers and articles, it is certainly worthwhile having yet another sceptical view to add to the information that veterinarians and others, including regulators, can use in an attempt to discredit or limit the use of ineffective therapies including acupuncture on animals.
There are a number of weaknesses. The main weakness is the lack of evidence or references to various statements. ie. ‘within the last few decades complementary and alternative medicine have gained popularity in the veterinary field’. ‘Although several textbooks....these efforts have not succeeded....’ and ‘veterinary acupuncture is a growing therapeutic field’.
Statements that CAVM have gained popularity are regularly made by proponents of acupuncture and TCVM but are they true? In addition, there are more resources than a few textbooks. The EBVMA skeptvet, Friends of Science in Medicine, Science-Based Medicine and others have regularly updated information on their webpages and publications. These bodies actively campaign against the use of CAVM. Maybe these do have an impact.
As a small animal practice owner and practitioner in an inner-city practice, I can report that very few of my clients enquire about the use of CAVM therapies and there is no increasing trend to do so.
The statement that the author doesn’t expect this paper to change the mind set of practitioners but hopes to do so to others new to the field and those contemplating its use could be tied in better with the statement about veterinary regulators as restrictions are a powerful tool in changing behaviours.
I believe there is some scope to mention that further discussion on the animal welfare and ethical impacts of the use of ineffective therapies, including acupuncture would be useful. There is just one sentence ‘This can endanger animal welfare and the public good and poses a serious reputational risk to the veterinary profession’ without supporting evidence.
In addition, I believe there is some scope to mention that further analysis and discussion on the significant animal welfare impacts of the use of TCM on other animals such as rhinos, bears, pangolins and donkeys may be worthwhile. Perhaps potential providers and users of CAVM may be persuaded by reference to animal welfare.
In the title the words ‘Traditional Veterinary Chinese Acupuncture’ are used. Yet they not used again in the text. These words are not in common usage as far as I’m aware. Throughout the article the words Traditional Chinese Veterinary Medicine (TCVM), or TCM, which are in common usage, are used. Section 4 uses the term ‘Traditional Chinese Veterinary Acupuncture’.
Commonly veterinary acupuncturists state that Veterinary Acupuncture is part of a whole system of TCVM and the term veterinary acupuncture is in common use.
Author Response
As a proponent of the use of EBVM and as a veterinarian concerned with the animal welfare and ethical issues associated with the use of CAVM I was pleased to read this paper that takes a critical and sceptical look at the use of veterinary acupuncture. I thought the paper achieved its aim and arguments were made in a logical and appropriate fashion. The paper maintained my interest.
The strength of this paper is that it adds to the literature on this topic and examines the basis of the belief in Traditional Chinese Veterinary Acupuncture and brings together various threads, historical and scientific regarding the use of acupuncture in animals. The arguments that Traditional Chinese Veterinary Medicine is based on pre-scientific principles and acupuncture is a placebo are well made and the reference list is extensive.
Although these views have been made many times before in textbooks, published papers and articles, it is certainly worthwhile having yet another sceptical view to add to the information that veterinarians and others, including regulators, can use in an attempt to discredit or limit the use of ineffective therapies including acupuncture on animals.
Thank you for your comments.
The paper is well structured and easy to read although some of the English needs tidying up.
English language was revised for accuracy and clarity.
There are a number of weaknesses. The main weakness is the lack of evidence or references to various statements. ie. ‘within the last few decades complementary and alternative medicine have gained popularity in the veterinary field’. ‘Although several textbooks....these efforts have not succeeded....’ and ‘veterinary acupuncture is a growing therapeutic field’.
I acknowledge the reviewer’s concerns that, taken from their original context, those sentences may be interpreted as supportive of CAM. On the one hand, I do not intend to give CAM more credit than it deserves but, on the other hand, I need to stress that CAM is enough of a relevant topic to justify my approach (instead of an obscure topic performed by only a few vets and with very little impact on the vet profession). Dozens of different techniques fall within the umbrella of acupuncture and it is hard to deny that acupuncture has been growing in diversity and complexity (with stem cell acupuncture as the epitome of ‘modern acupuncture’). Please refer to the following comments to see how these concerns were dealt.
Statements that CAVM have gained popularity are regularly made by proponents of acupuncture and TCVM but are they true?
I think they are true, at least regarding TCM. China has considered Portugal as a strategic gateway to introduce TCM in Europe due to the long-standing cultural exchanges between the two countries. In 2018 alone, two professional organisations devoted to veterinary acupuncture have been established in Portugal (a statement now included in the manuscript, line 45). Every major University in Portugal has now a Confucius Institute, supported by the Chinese Government, and similar Institutes have been created throughout Europe, the US and Canada (https://en.wikipedia.org/wiki/Criticism_of_Confucius_Institutes). Furthermore, I also think that TCM works as a magnet for other CAMs (a claim that warrants further anaysis). The increasing support of TCM by the WHO and the risks it entails (cf. https://www.economist.com/news/china/21727945-unproven-remedies-promoted-state-why-chinas-traditional-medicine-boom-dangerous) urge us, more than ever, to voice our scepticism.
In addition, there are more resources than a few textbooks. The EBVMA skeptvet, Friends of Science in Medicine, Science-Based Medicine and others have regularly updated information on their webpages and publications. These bodies actively campaign against the use of CAVM. Maybe these do have an impact.
In accordance to your comments, I have changed the abstract (‘many authors’ instead of ‘several textbooks’) and the Introduction (including reference to internet fora such as The Skeptvet).
As a small animal practice owner and practitioner in an inner-city practice, I can report that very few of my clients enquire about the use of CAVM therapies and there is no increasing trend to do so.
The reviewer is most likely right, and clients are probably not to blame. I think that vets themselves are creating a demand for CAM, a claim that is made towards the end of the conceptual/historical critique, and that is further developed in the last paragraph of the paper. I have reinforced that argument with the concept of ‘collective amnesia’, that prevents us from seeing the resemblances between Qi and Pneuma and between acupuncture and bloodletting. (line 318)
The statement that the author doesn’t expect this paper to change the mind set of practitioners but hopes to do so to others new to the field and those contemplating its use could be tied in better with the statement about veterinary regulators as restrictions are a powerful tool in changing behaviours.
One of the things that I wanted to avoid is being accused of dogmatism, i.e. of someone who a priori rejects any kind of CAM. After such a critical paper, I felt that I needed a concluding paragraph that reached those in doubt, offering them a conciliatory and self-empowered solution. I would prefer to keep this paragraph separate from regulatory restrictions. Nonetheless, a sentence reinforcing restrictions was added to the previous paragraph.
I believe there is some scope to mention that further discussion on the animal welfare and ethical impacts of the use of ineffective therapies, including acupuncture would be useful. There is just one sentence ‘This can endanger animal welfare and the public good and poses a serious reputational risk to the veterinary profession’ without supporting evidence.
Thank you for pointing this out. This paragraph is intended to set the scene for another manuscript (in preparation) dealing with the regulatory, ethical and welfare issues of CAM. Nevertheless, a supporting reference was added.
In addition, I believe there is some scope to mention that further analysis and discussion on the significant animal welfare impacts of the use of TCM on other animals such as rhinos, bears, pangolins and donkeys may be worthwhile. Perhaps potential providers and users of CAVM may be persuaded by reference to animal welfare.
I appreciate this comment but, as mentioned in the introduction, I do not which to deviate from the focus of the paper, acupuncture, and move towards Herbal Medicine (the culprit of those animal welfare issues). This has turned into a long paper and any consideration regarding Herbal Medicine would seem superficial. Again, a paper on the topic of animal welfare and ethics is currently being prepared.
In the title the words ‘Traditional Veterinary Chinese Acupuncture’ are used. Yet they not used again in the text. These words are not in common usage as far as I’m aware. Throughout the article the words Traditional Chinese Veterinary Medicine (TCVM), or TCM, which are in common usage, are used. Section 4 uses the term ‘Traditional Chinese Veterinary Acupuncture’. Commonly veterinary acupuncturists state that Veterinary Acupuncture is part of a whole system of TCVM and the term veterinary acupuncture is in common use.
The term ‘Traditional Chinese Veterinary Acupuncture’ [and not ‘Traditional Veterinary Chinese Acupuncture’, my mistake!] used in the title was intended to encapsulate both ‘Traditional Chinese Veterinary Medicine’ and ‘Veterinary Acupuncture’. I am aware that this is not an established term and that is why it was only used once within the text. However, it is both short and comprehensive. Several alternatives were considered, namely:
a) The emperor’s new clothes - an epistemological critique of Veterinary Acupuncture
b) The emperor’s new clothes - an epistemological critique of Traditional Chinese Veterinary Medicine and Acupuncture.
c) The emperor’s new clothes - an epistemological critique of Traditional Chinese Medicine and Veterinary Acupuncture.
Reluctance to use a) was due to the fact that most of the conceptual critique is aimed at Traditional Chinese (Veterinary) Medicine and it is not self-explainable for a lay reader that veterinary acupuncture is part of TCVM. Alternatives b) and c) may give the false impression that TCM and Acupuncture will be dealt separately or that Medicine (meaning remedies) is also part of the analysis. Taking all these reflections in consideration, I opted for ‘Traditional Veterinary Chinese Acupuncture’. I am, however, willing to reconsider.
ADDITIONAL CORRECTIONS (track changes)
Line 39 - I have clarified that the term non-conventional therapies will be used to designate complementary and alternative medicines.
Line 256 - Regarding the first use of European acupuncture, a sentence was changed because I found a reference showing that it was the editor of the French scientific journal Recueil de Médecine Vétérinaire who, excited by experiments in humans, encouraged veterinary practitioners to apply acupuncture in animals.
Line 295 - The role of Thomas Kuhn in the development of my own argument is further clarified.
Line 348 - A sentence was added explaining that the study originally aimed at comparing the electroencephalographic recordings of dogs suffering from uncontrolled seizures.
Line 345 - A sentence was added briefly explaining the findings of the paper.
Line 411 - A paragraph describing a randomised, controlled clinical trial aimed to assess the efficacy of dry needle acupuncture to treat dog hip dysplasia was added. The study was published in JAVMA in 2016 and has led to an interesting debate regarding its validity.
Reviewer 2 Report
The manuscript is very well written and based on a thorough exploration of the history of traditional veterinary practices in Europe and China. The subject is of high importance to the practice of veterinary medicine, relating to the widespread misapprehension that modern scientific medicine is traditional European medicine, when in fact most traditional European medical beliefs have been abandoned as a result of scientific evidence of their futility. This misunderstanding has led many veterinarians in the west to assume that it is inappropriate to apply the scientific method to other traditional forms of medicine, impeding the critical assessment of non-European historical practices. This manuscript will help readers to appreciate the fundamental similarities between traditional European and Chinese medicine, which should make apparent the applicability of the scientific method to the assessment of medical practices from any culture.
Minor edits
Line 87: I believe "acupoints" should be used in place of "meridians"
Line 89: Delete the second appearance of the word "and"
Line 341: "Spurious" would convey the meaning more accurately than "serendipitous"
Line 357: "Dodged" is unnecessarily emotionally laden; perhaps simply say "avoided"
Line 375: replace "worlds" with "words"
Line 400: correct spelling of "analysing"
Line 424: "...either form of self-healing is independent..."
Line 442: I suggest the word "critical" be used instead of "sceptic[al]," since in colloquial English "sceptic" has the connotation of "cynical"
Line 483: "...it be conceptual"
Minor comment
The analogy presented in lines 432 to 436 is somewhat weakened by the fact that much of the placebo effect in veterinary medicine is simply a result of the owner or veterinarian mistaking natural healing for effects of the intervention. It might be more helpful to readers to present an analogy illustrating the thought processes that might have become widespread had the Chinese, rather than Europeans, been the first to examine their traditional medical practices scientifically. In such a scenario, having abandoned and forgotten most of their historical beliefs, and erroneously viewing scientific medicine as TCM, those in the East could be arguing for the acceptance of traditional European bloodletting and associated practices, and employing them alongside modern conventional practices, arguing that it would be wrong to use science to assess anything but Chinese medicines. I think that would go a long way to illustrating one of the fundamental errors the authors raise.
Author Response
The manuscript is very well written and based on a thorough exploration of the history of traditional veterinary practices in Europe and China. The subject is of high importance to the practice of veterinary medicine, relating to the widespread misapprehension that modern scientific medicine is traditional European medicine, when in fact most traditional European medical beliefs have been abandoned as a result of scientific evidence of their futility. This misunderstanding has led many veterinarians in the west to assume that it is inappropriate to apply the scientific method to other traditional forms of medicine, impeding the critical assessment of non-European historical practices. This manuscript will help readers to appreciate the fundamental similarities between traditional European and Chinese medicine, which should make apparent the applicability of the scientific method to the assessment of medical practices from any culture.
Thank you for your comments.
Minor edits
Line 87: I believe "acupoints" should be used in place of "meridians"
Done as suggested
Line 89: Delete the second appearance of the word "and"
Done as suggested
Line 341: "Spurious" would convey the meaning more accurately than "serendipitous"
Done as suggested
Line 357: "Dodged" is unnecessarily emotionally laden; perhaps simply say "avoided"
Done as suggested
Line 375: replace "worlds" with "words"
Done as suggested
Line 400: correct spelling of "analysing"
Done as suggested
Line 424: "...either form of self-healing is independent..."
Done as suggested
Line 442: I suggest the word "critical" be used instead of "sceptic[al]," since in colloquial English "sceptic" has the connotation of "cynical"
Done as suggested
Line 483: "...it be conceptual"
Done as suggested
Minor comment
The analogy presented in lines 432 to 436 is somewhat weakened by the fact that much of the placebo effect in veterinary medicine is simply a result of the owner or veterinarian mistaking natural healing for effects of the intervention. It might be more helpful to readers to present an analogy illustrating the thought processes that might have become widespread had the Chinese, rather than Europeans, been the first to examine their traditional medical practices scientifically. In such a scenario, having abandoned and forgotten most of their historical beliefs, and erroneously viewing scientific medicine as TCM, those in the East could be arguing for the acceptance of traditional European bloodletting and associated practices, and employing them alongside modern conventional practices, arguing that it would be wrong to use science to assess anything but Chinese medicines. I think that would go a long way to illustrating one of the fundamental errors the authors raise.
I understand the reviewer’s standpoint and gave it considerable thought. I think that I have mostly addressed the concern with the analogy when considering that cure can be a result of the natural course of illness. Moreover, the second part of this comment brings us to what has become known as Needham question: “Why china’s civilization stopped developing in the 16th century?” The plausibility of the reviewer’s suggestion can only be understood after comparing European and Chinese historical, scientific and philosophical contexts. In the words on Joseph Needham himself,
Why did modern science, the mathematization of hypotheses about Nature, with all its implications for advanced technology, take its meteoric rise only in the West at the time of Galileo [but] had not developed in Chinese civilisation or Indian civilisation? Gunpowder, the magnetic compass, and paper and printing, which Francis Bacon considered as the four most important inventions facilitating the West's transformation from the Dark Ages to the modern world, were invented in China".
Honestly, I am not sure I would be able to flip the coin and address this topic in a meaningful way in the present manuscript.
ADDITIONAL CORRECTIONS (track changes)
Line 39 - I have clarified that the term non-conventional therapies will be used to designate complementary and alternative medicines.
Line 256 - Regarding the first use of European acupuncture, a sentence was changed because I found a reference showing that it was the editor of the French scientific journal Recueil de Médecine Vétérinaire who, excited by experiments in humans, encouraged veterinary practitioners to apply acupuncture in animals.
Line 295 - The role of Thomas Kuhn in the development of my own argument is further clarified.
Line 348 - A sentence was added explaining that the study originally aimed at comparing the electroencephalographic recordings of dogs suffering from uncontrolled seizures.
Line 345 - A sentence was added briefly explaining the findings of the paper.
Line 411 - A paragraph describing a randomised, controlled clinical trial aimed to assess the efficacy of dry needle acupuncture to treat dog hip dysplasia was added. The study was published in JAVMA in 2016 and has led to an interesting debate regarding its validity.
Reviewer 3 Report
This paper is based on insufficient arguments and has a shallow understanding of acupuncture and moxibustion in traditional Chinese medicine. It is suggested that the historical origin and essence of acupuncture and moxibustion should be further understood.
The author compares acupuncture and moxibustion in ancient China with bloodletting to show that acupuncture and moxibustion are similar to bloodletting. However, modern acupuncture and moxibustion are quite different from ancient acupuncture and moxibustion, and the examples cited by the author are not enough to prove that acupuncture and moxibustion are similar to bloodletting.
Author Response
Point 1: This paper is based on insufficient arguments and has a shallow understanding of acupuncture and moxibustion in traditional Chinese medicine. It is suggested that the historical origin and essence of acupuncture and moxibustion should be further understood.
Response 1: I thank the reviewer for disagreeing with the views espoused in the manuscript, in which lies the foundations for scientific debate. However, the reviewer has failed to present one single argument that may demonstrate my “shallow understanding of acupuncture and moxibustion in traditional Chinese medicine”.
Point 2: The author compares acupuncture and moxibustion in ancient China with bloodletting to show that acupuncture and moxibustion are similar to bloodletting. However, modern acupuncture and moxibustion are quite different from ancient acupuncture and moxibustion, and the examples cited by the author are not enough to prove that acupuncture and moxibustion are similar to bloodletting.
Response 2: I agree with the reviewer’s remark that modern acupuncture is quite different from ancient acupuncture. In effect, the manuscript presents a rationale that helps explaining why needling in China has evolved from letting blood to fine needle acupuncture. Furthermore, the reviewer considers that not enough has been done to prove that acupuncture and moxibustion are similar to bloodletting. However, no arguments or peer-reviewed references are presented. Nonetheless, in order to address the reviewer’s concerns, the similarities between bloodletting and acupuncture, and between Humoral Medicine and TCM have been reinforced as follows:
Figure 1 - A third image was included, supporting the claim that any of the nine ancient Chinese needles can be considered similar to ancient surgical instruments.
Figure 2 - A third image was included, reinforcing the similarities between the two illustrations. The description of Figure 1a was slightly corrected.
ADDITIONAL CORRECTIONS (track changes)
Line 39 - I have clarified that the term non-conventional therapies will be used to designate complementary and alternative medicines.
Line 256 - Regarding the first use of European acupuncture, a sentence was changed because I found a reference showing that it was the editor of the French scientific journal Recueil de Médecine Vétérinaire who, excited by experiments in humans, encouraged veterinary practitioners to apply acupuncture in animals.
Line 295 - The role of Thomas Kuhn in the development of my own argument is further clarified.
Line 348 - A sentence was added explaining that the study originally aimed at comparing the electroencephalographic recordings of dogs suffering from uncontrolled seizures.
Line 345 - A sentence was added briefly explaining the findings of the paper.
Line 411 - A paragraph describing a randomised, controlled clinical trial aimed to assess the efficacy of dry needle acupuncture to treat dog hip dysplasia was added. The study was published in JAVMA in 2016 and has led to an interesting debate regarding its validity.